# MixFormer: A Self-Attentive Convolutional Network for 3D Mesh Object Recognition

**Lingfeng Huang, Jieyu Zhao \* and Yu Chen**

Mobile Network Application Technology Laboratory, School of Information Science and Engineering, Ningbo University, 818 Fenghua Road, Ningbo 315211, China
\* Correspondence: zhao_jieyu@nbu.edu.cn

**Abstract:** 3D mesh as a complex data structure can provide effective shape representation for 3D objects, but due to the irregularity and disorder of the mesh data, it is difficult for convolutional neural networks to be directly applied to 3D mesh data processing. At the same time, the extensive use of convolutional kernels and pooling layers focusing on local features can cause the loss of spatial information and dependencies of low-level features. In this paper, we propose a self-attentive convolutional network MixFormer applied to 3D mesh models. By defining 3D convolutional kernels and vector self-attention mechanisms applicable to 3D mesh models, our neural network is able to learn 3D mesh model features. Combining the features of convolutional networks and transformer networks, the network can focus on both local detail features and long-range dependencies between features, thus achieving good learning results without stacking multiple layers and saving arithmetic overhead compared to pure transformer architectures. We conduct classification and semantic segmentation experiments on SHREC15, SCAPE, FAUST, MIT, and Adobe Fuse datasets. Experimental results show that the network can achieve 96.7% classification and better segmentation results by using fewer parameters and network layers.

**Keywords:** transformer; 3D convolutional network; 3D object recognition; vector self-attention; 3D model segmentation





## 1. Introduction

Compared with 2D data, 3D data contains rich spatial information and target details, providing the possibility to achieve more detailed computer vision tasks. At the same time, with the development of 3D vision technology, the cost of acquiring and processing 3D data is getting lower and lower, thus giving rise to emerging application areas such as autonomous driving [1–3], augmented reality [4,5], and robotics. In recent years, how to process 3D data with the help of deep learning methods [6–8], which are more mature in the field of 2D image research [9–14], has received a great deal of attention from scholars. However, unlike a 2D image where pixels are uniformly distributed on a 2D grid, a 3D grid is a collection of connected relationships between a series of points and edges in 3D space. This means that the 3D grid model does not have a regular 3D spatial representation, while the input of data is disordered. The structural differences between 2D images and 3D grid models make it impossible to directly apply the more mature neural network design in the field of 2D image research to 3D mesh models.

To overcome the problem of irregularity and disorder in 3D grid models, some studies have pioneered attempts by designing the input format of 3D grid model data and 3D convolution and 3D pooling methods [15–18]. Feng et al. and Hanocka et al. [16,17] used grid or grid edge as the basic unit to define the standard input format of 3D grid models and, based on the input features, for 3D grid models design convolutional classification networks based on the input features. However, although the convolution and pooling process can weigh the combination of lower-level features to form higher-level features,

the spatial information and dependencies of the lower-level features in the higher-level features are lost [19]. Especially in 3D space, the 3D model is richer in location structure information and the local–local interrelationships are closer.

To preserve inter-feature dependencies, inspired by the self-attentive mechanisms that have been successful in the natural language domain and in the 2D image domain, work such as point cloud transformer (PCT) [20], point transformer (PT) [21], and transformer (METRO) [22] in a 3D point cloud model analysis using transformer architecture. However, these methods applied to the 3D domain encode the input in a way that does not design a self-attentive mechanism based on the local features of the 3D model and are not as interpretable as tokens in NLP. At the same time, such transformer networks, which are stripped from the original convolutional architecture, tend to retain the multi-layered characteristics of convolutional networks, which often implies a huge computational overhead when combined with the self-attentive mechanism with $O(n^2)$ time complexity in a transformer [23].

Based on the characteristics of convolutional networks and transformer networks, this paper proposes a self-attentive convolutional network MixFormer for 3D mesh models. The shape features of 3D mesh models are extracted as the semantic information input to the transformer module through the front 3D mesh convolutional module. To ensure that the 3D mesh convolution module can effectively extract the feature information of the 3D mesh model, a 3D convolution kernel is defined in this paper. The introduction of the 3D mesh transformer module enables the network to learn the global association among the 3D mesh shape features, which makes up for the deficiency of the convolution operation in learning long-distance features. And by introducing a vector-type self-noticing mechanism in the transformer network, the spatial information of the 3D grid model is introduced, which makes the transformer module better applicable to the 3D grid model.

In this paper, we conducted classification experiments on SHREC15 and Manifold40 datasets, and segmentation experiments on SCAPE, FAUST, and MIT datasets, and demonstrated the effectiveness of each module on 3D feature learning by ablation experiments. The experiments show that MixFormer can achieve good learning results without stacking multiple layers and with less computational overhead by learning local feature information through the front 3D grid convolution module and establishing dependencies between features by the 3D grid transformer.

In summary, the main contributions of this paper are as follows.

(1) A 3D convolutional kernel applicable to 3D mesh models is designed to enable the network to extract feature information on 3D grid models with irregularities and disorder.

(2) The vector-based transformer module is designed to better learn the dependencies between features through a vector-based self-attentive mechanism and a learnable position encoding.

(3) The shape features extracted by the convolutional network are processed by using the self-attentive mechanism with a pyramidal structure, so that the network can fully establish global feature dependencies while extracting feature information more accurately.

## 2. Related Work

### 2.1. 3D Convolutional Network

The mesh of a 3D mesh model consists of a connection relation between vertices and edges with irregular distributions, so it is not possible to use the 2D convolution method directly on the 3D mesh model, which means that a suitable 3D convolution operation needs to be defined for the structure of the 3D mesh model. Feng et al. proposed MeshNet [16], which firstly, defines an input format for the mesh: for each mesh, extract its centroid, the centroid-to-vertex vector, the normal vector of the mesh, and the neighboring mesh coordinates as its input features, and then design the CNN classification network. Hanocka et al. proposed MeshCNN [17], which defines the convolutional neighborhood

in terms of edges and defines a formula to transform the four edges of the neighborhood to ensure the invariance of the convolutional operation. Moreover, the dihedral angle, the interior angle of two faces, and the edge length ratio of two faces above the base are used as five-dimensional input features, and the pooling is defined by edge folding in the pooling operation.

The above method follows the 2D convolutional approach in processing 3D mesh models by aggregating local features captured by the convolutional kernel through pooling operations, thus facilitating the extraction of high-level features by the posterior network [12]. Although convolutional operations can effectively capture local information, vision tasks such as object detection [24–29], instance segmentation [30–32], and key-point detection often require the establishment of long-range dependencies [33], and convolution-based architectures often require stacking multiple layers in order to aggregate local features and improve the performance of convolutional backbone networks [10,34]. However, although the convolutional, pooling process can form higher-level features by weighting the combination of lower-level features, it loses the spatial information and dependencies of the lower-level features in the higher-level features [19]. Three-dimensional grid models form spatial surface features through the connection relationship of points and edges, and the loss of spatial location information often has a significant impact on the extraction of three-dimensional target recognition [12]. For example, in a human model, two meshes that are similar in spatial location may be distributed on different fingers, and if we only focus on the local area, we lose its self-contained semantic information. Therefore, a mechanism for modeling based on global (non-local) dependencies may be a more robust and scalable solution.

### 2.2. Transformer

Establishing long-range dependencies is not only important for 3D target feature learning, but also for Natural Language Processing (NLP). In recent years, the transformer has become increasingly popular in NLP based on the matching mechanism and parallelizability of the encoding. This has now become a standard tool in NLP in the form of a transformer [35], with prominent examples being the GPT [36,37] and BERT [38,39] models.

Noting the excellent ability of the self-attention to establish long-range dependencies, several research workers have tried to apply the self-attention to the 2D image domain and the three-dimensional vision domain. In the 2D image domain, a simple way to use the self-attention is to replace the convolutional layer with the multi-head self-attention (MHSA) layer proposed in the transformer [35]. SASA [40], AACN [41], SANet [42], Axial-SASA [43], etc., introduce various forms of a transformer (local, global, axial, vector) by replacing the original convolutional layers in the ResNet [10] backbone network. On the other hand, methods such as vision transformer (VIT) [44] and Swin transformer [45] segment images into non-overlapping and overlapping blocks, and then by linearly stacking transformer blocks, while ensuring the interpretability of the input tokens, the feature learning of the image is achieved. In the field of 3D vision, a point cloud transformer (PCT) [20] proposes a point-based transformer, which learns features through a vector representation of the transformer. Point transformer (PT) [21] enhances the potential feature representation of the input to better capture the local features in the point cloud. local features. A mesh transformer (METRO) [22], on the other hand, performs human shape reconstruction on 3D mesh models. However, none of these methods applied to the 3D domain encodes the input in a way that is as interpretable as token in NLP. At the same time, such transformer networks derived from the original convolutional architecture tend to retain the multi-layered nature of convolutional networks, which often implies a huge computational overhead when combined with the self-attention of $O(n^2)$ time complexity in a transformer [23].

Inspired by the above work, in order to better establish the global dependency of feature information in the network and at the same time reduce the number of network parameters so that the network can be better applied to 3D mesh models, three innova-

tions are proposed in this paper: (1) a 3D convolutional kernel is designed for 3D mesh models, which enables the network to extract feature information on 3D mesh models with complexity and disorder; (2) a vector-type transformer block is designed to better learn the dependencies among the features through the vector-based self-attention and the learnable position coding; (3) the shape features extracted by the convolutional network are processed by the self-attention with the pyramidal structure so that the network can fully establish the global feature dependencies while extracting the feature information more accurately.

## 3. MixFormer

In this paper, we propose a vector self-attention convolutional network, MixFormer, applied to 3D mesh models from the perspective of local feature learning to establish global feature dependencies. The specific network model design is shown in Figure 1, which can be divided into the following two major blocks.

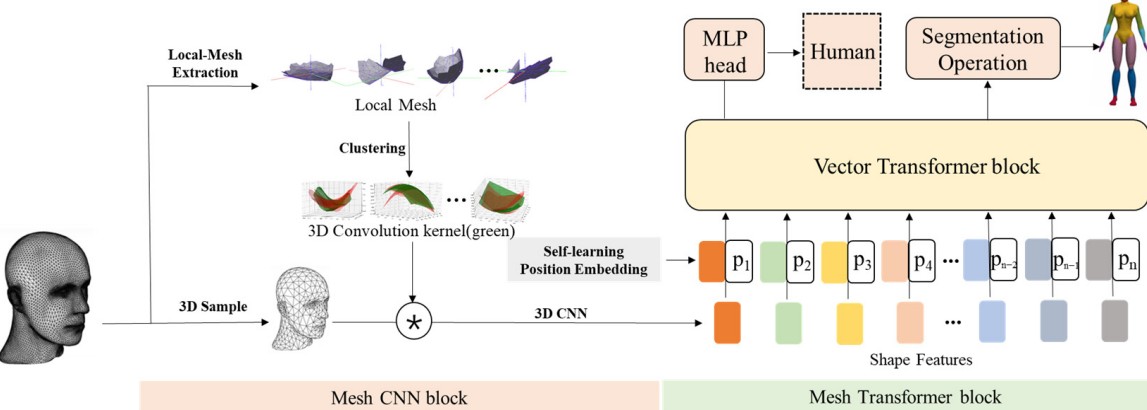

**Figure 1.** MixFormer network structure and 3D target processing flow. MixFormer can be viewed as two blocks: a mesh CNN block and mesh transformer block. A mesh CNN block can acquire local surface features of the 3D mesh model and use them as Token input to the mesh transformer block. The block learns global features by building dependencies between local surface features. The learned features can be used for downstream tasks such as classification and segmentation.

1.  3D mesh convolution block: For the input 3D mesh data, the 3D mesh convolution block defines the local surface by polynomial expression, and acquires the 3D convolution kernel by aggregating the local surface features through clustering operation. Then the data is reduced by 3D sampling to get the shape feature representation of 3D mesh model data.
2.  3D mesh transformer block: for the shape feature representation of the 3D mesh model data obtained after processing by the convolution block, the spatial feature information of the 3D mesh model is introduced through the learnable position encoding, and the global dependency of its high-level semantic features is established using the vector self-attention, so as to realize the downstream tasks such as classification and semantic segmentation of the 3D mesh models.

Compared with previous methods, this method has the following advantages: (1) A learnable 3D convolutional kernel template is introduced to cope with the problem that traditional convolutional operations are difficult to apply to 3D mesh models. (2) The transformer block, which is suitable for establishing long-distance dependencies, is introduced to learn the association between global features and make up for the deficiency of the convolution block in learning long-distance features. (3) The combination of the convolution block and the transformer block, which learns the shape features in the 3D mesh model through the pre-convolution operation and then establishes the association between

the features through the transformer block, makes MixFormer achieve good classification and segmentation results without building a multi-level network architecture. Compared with other network models applied to 3D mesh models, the number of parameters and computational effort is significantly reduced.

*3.1. 3D Mesh Convolution Block*

To perform convolutional operations on surface features of complex 3D mesh models, a polynomial convolutional kernel is designed in this paper. The method is described as follows.

Taking the vertices of the 3D mesh model as the center of the window, we first perform a breadth-first search on each vertex to obtain its K-neighborhood, and call the 3D mesh surface formed by the vertices and interconnected edges contained in its K-neighborhood a local surface window. For the local surface window, we define the polynomial representation equation of the local surface and fit the surface by weight learning to obtain the polynomial expression of the local surface. The obtained polynomials for local surfaces are then clustered to extract the 25 surface shapes commonly found in the model as the base convolution kernels. The similarity of each local surface to the basic surface convolution kernel is obtained by convolving each local surface using the basic surface convolution kernel. This is used as the local surface feature of the vertex neighborhood.

3.1.1. Mesh Local Surface Representation

First of all, since the same model in the 3D model undergoes translation or rotation, its polynomial expression parameters are changed accordingly. In order to eliminate the effect of translation and rotation on the features, the grid window needs to be orthogonalized so that its window center is the origin of the new coordinate system (translation) and its normal vector is the *z*-axis of the new coordinate system (rotation).

Second, for each vertex's K-neighborhood, we have the following definition.

$$V_{\text{win}-i} = \left\{ v_j \in V_{i-nei}, j = 1, 2, \cdots, K-1 \right\} \tag{1}$$

$$E_{\text{win}-i} = \left\{ (v_a, v_b) \mid v_a, v_b \in V_{\text{win}-i} \right\} \tag{2}$$

Here $V_{i-nei}$ denotes the set of vertices contained in the K-neighborhood of vertex $v_i$, $V_{\text{win}-i}$ is the set of vertices contained in the window with vertex $v_i$ as the window center, and $E_{\text{win}-i}$ is the set of edges contained in the window with vertex $v_i$ as the window center.

Since the local window contains only a small amount of 3D surface mesh when the local window is designed small, the ability to aggregate its neighborhood information is limited. Moreover, when the window is designed larger, the window shape becomes complex, and the original 3D coordinates of the vertices alone are not sufficient to describe the mesh features. Therefore, geodesic distances are introduced in related works. However, the calculation of geodesic distance often requires a lot of computational resources, so block distance d is chosen as the approximation of geodesic distance.

Therefore, the fitting function of the local window is shown in Equation (3).

$$F(v_c \mid \theta) = z - \left( \theta_0 + \theta_1 x + \theta_2 y + \theta_3 d + \theta_4 x^2 + \theta_5 y^2 + \theta_6 d^2 + \theta_7 xy + \theta_8 xd + \theta_9 yd \right) \tag{3}$$

Let the fitting function $F(v_c \mid \theta)$ of the local window surface equal to 0, train the learnable parameters $\theta_i$ therein, and fit the local surface polynomial. The function $F(v_c \mid \theta)$ obtained after fitting is the approximate representation of the local window.

In the training process, the loss function is the mean loss:

$$L_f = \frac{1}{K} \sum_{v}^{V} F(v_c \mid \theta) \tag{4}$$

In order to obtain the basis surfaces in the model as the base convolution kernel, the obtained local window expression features need to be clustered. In this paper, it is divided into 25 types of basis surfaces. The feature representation of the base surface obtained by clustering is the base convolution kernel. Finally, we use the base convolution kernel to convolve each positive-definite local surface to obtain the similarity between the local surface and the base surface as its shape features.

### 3.1.2. Similarity Measure

In order to perform the clustering operation on the local window, the similarity definition of the local surface polynomials is needed to evaluate the clustering effect. To define the similarity between local surfaces, in this paper, the average distance between surfaces is used as a measure of the similarity between surfaces. For this purpose, combined with Equation (3), we have the following equation:

$$\mathrm{Dist}(S_i, S_t) = \sum_{v_j \in S_i} |F(v_j \mid \theta_t)| \tag{5}$$

where the surface polynomials $F(v_i \mid \theta)$ and $F(v_t \mid \theta)$ corresponding to the local surface $S_i$ and the target surface $S_t$ (the center of clustering), respectively. $|F(v_j \mid \theta_t)|$ denotes the distance between the points of the local surface $S_i$ and the target surface $S_t$. Since Equation (5) only represents the one-way distance, to reduce loss, we define a two-way difference metric based on Equation (5).

$$\mathrm{DIF}(S_i, S_t) = \frac{1}{2}(\mathrm{Dist}(S_i, S_t) + \mathrm{Dist}(S_t, S_i)) \tag{6}$$

For each local spatial surface of the 3D mesh model, the polynomial representation function is $F(v_c \mid \theta)$. We assume that the local space surfaces in the 3D mesh model obey a Gaussian distribution on the data with variance $\sigma$ and mean $\mu$. Then, for any local space surface $S_i$, the probability that the surface belongs to the target surface is Equation (7).

$$P(S_i \mid \theta, \sigma) = \frac{1}{\sqrt{2\pi\sigma^2}} \exp\left( -\frac{\sum_{v_i \in S_i} (F_i(v_i|\theta) - \mu)^2}{2\sigma^2} \right) \tag{7}$$

### 3.1.3. Definition of 3D Convolution Operations

In the convolution operation of 2D images, the 2D convolution kernel convolves the local image, the essence of which can be understood as computing the similarity between the 2D convolution kernel and the local image. In Section 3.1.2, we define the similarity function $P(S_i \mid \theta, \sigma)$, which is used to perform the clustering operation on the local surface windows to obtain the standard surface windows (clustering centers). The standard surface window obtained by the clustering operation can be used as a 3D convolution kernel for the 3D convolution operation. The 3D convolution operation is the process of calculating the 3D convolution kernel and the local surface window, so the similarity measure can still be used in Equation (7).

To make the similarity function $P(S_i \mid \theta, \sigma)$ closer to the common convolution operation, taking the logarithm of the full probability equation yields.

$$\ln P(S_i \mid \theta, \sigma) = -\frac{1}{2\sigma^2} \times \sum_{v_i \in S} 1 \times (F(v_i|\theta) - \mu)^2 - \frac{1}{2}\ln\sqrt{2\pi\sigma^2} \tag{8}$$

This formula can be converted into a convolution formula:

$$\mathbf{Y} = \mathbf{WX} + \mathbf{b} \tag{9}$$

$\mathbf{X}$ represents the feature vector $(x_1, x_2, \ldots, x_k)$ and $x_i$ represents the feature at vertex $v_i$. For example 0, 1 can represent the presence or absence of vertex $v_i$ on a local surface, where

**W** represents the weight vector $(w_1, w_2, \ldots, w_k)$, and b represents the bias corresponding to the feature vector.

$$w_i = -\frac{1}{2\sigma^2} \times \sum_{v_i \in S} 1 \times (F(v_i|\theta) - \mu)^2 \tag{10}$$

$$b = -\frac{1}{2}\ln\sqrt{2\pi\sigma^2} \tag{11}$$

3.1.4. Three-Dimensional Sampling

After the convolution operation by the polynomial convolution kernel, each vertex aggregates the local surface features of its neighborhood. However, this introduces a new problem, as each vertex contains the local surface features of its neighborhood, so there is a large amount of data redundancy. To reduce the amount of data in the network, 3D sampling can be performed based on the vertices.

Unlike the 3D point cloud models, the 3D mesh models represent the contour features of the model by the combination of vertices and triangulated facets, and the vertices and triangulated facets are also not uniformly distributed on the model, showing a partly dense and partly sparse feature. Therefore, if the vertices are to be sampled, the farthest sampling method, which is commonly used in point cloud models to cover all points in space as uniformly as possible, is not very suitable for 3D mesh models. To solve this problem, we found that Poisson disk sampling can achieve uniform sampling according to the model contour.

The algorithm for Poisson disk sampling is shown in Algorithm 1.

---

**Algorithm 1:** Poisson disk sampling

---

**Input:** Input vector $x_i$, desired number of samples $N$
**Output:** Output vector $y_j$

1.  Build a kd tree for samples
2.  Allocate a heap $S_i$ for each sample to store the weights $w_i$
3.  Assign initialized weights to each sample. $w_i = \sum_j w_{ij}$

$$w_{ij} = \left(1 - \frac{\hat{d}_{ij}}{2\sqrt[3]{\frac{A_3}{4\sqrt{2}N}}}\right)^8$$

4.  While number of samples $>$ desired :
    $S_i \leftarrow$ pull the top sample from heap
    For each sample $S_i$ around $S_j$
        Remove $w_{ij}$ from $w_i$
        Update the heap position of $w_i$

---

*3.2. 3D Mesh Transformer Block*

Through the 3D convolution operation, we extract the local shape features of the 3D mesh model. In order to establish the dependencies between the features, the local shape features of the 3D mesh model need to be input into the 3D mesh transformer block. The core structure of the 3D mesh transformer block is shown in Figure 2.

Considering the disorder of data in the 3D mesh model, i.e., the model features of the vertices in the 3D mesh are independent of the input order, an MLP network with shared weights is introduced in the first layer of the network to eliminate the effect of input order and extract the 3D mesh model features, feature mapping is performed for local shape features, and then the 3D mesh is mapped through multiple transformer layer and down layers to The global features of the model are then learned through multiple transformer layers and down layers. Finally, the features of the model are aggregated by a global maximum pooling operation and the model type is predicted by an MLP layer. For semantic segmentation operation, the transformer up layer is introduced to restore features to predict the semantic labels of mesh. The internal implementation of the transformer layer, down layer, and up layer is shown in Figure 3.

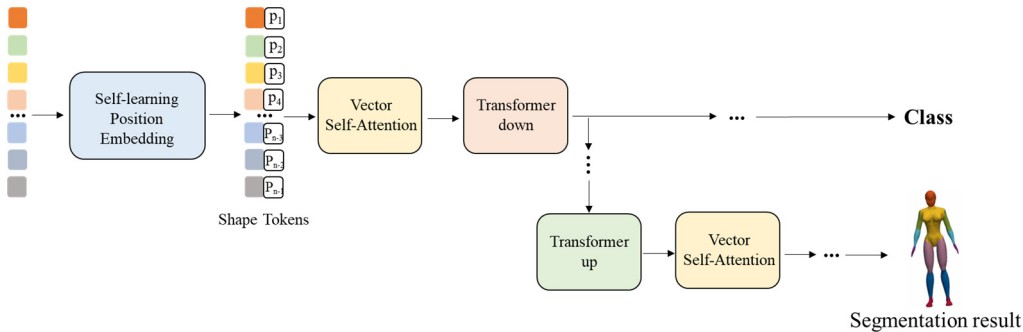

**Figure 2.** 3D mesh transformer block.

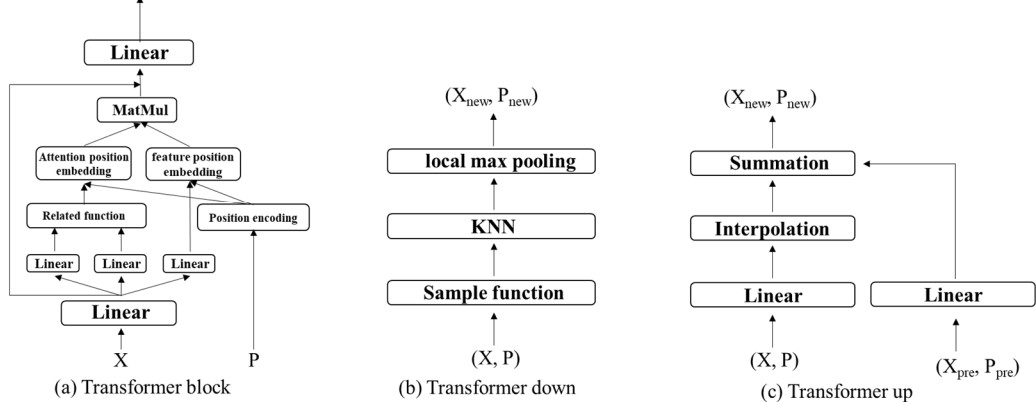

**Figure 3.** Internal implementation of transformer layer, transformer down layer, and transformer up layer.

### 3.2.1. Vector Self-Attention

The core of the model is the transformer layer containing residual blocks. vector self-attentiveness enables the exchange between local features and the acquisition of associations between global features. By linear mapping, the dimensionality of the data computed by the vector-based self-attention can be reduced.

The input of the traditional transformer's self-attention contains three matrices, $Q \in \mathbb{R}^{n \times D_k}$, $K \in \mathbb{R}^{m \times D_k}$, and $V \in \mathbb{R}^{m \times D_v}$. $Q, K, V'$ represent query, key, and value. $K$, $V'$ is like a key–value relationship, which is a one-to-one correspondence.

$$\text{Attention}(Q, K, V) = \text{softmax}\left( QK^\top + pos \right) V' \tag{12}$$

Self-attentions can be divided into two categories: scalar self-attentions and vector self-attentions.

The scalar self-attention can be expressed as Equation (13).

$$o_i = \sum_{\mathbf{x}_i \in \mathcal{X}} \psi\left( (q_i)^\top k_j + pos \right) v'_j \tag{13}$$

Corresponding to each input feature $x_i$, $o_i$ is the output feature. $\psi$ is a normalization function, such as softmax. $pos$ is a position encoding function. $q, k$, and $v$ represent the query, key, and value generated by the corresponding $x_i$. It can be expressed as Equation (14).

$$q_i = Q(x_i) \qquad k_i = K(x_i) \qquad v'_i = V'(x_i) \tag{14}$$

$Q, K$, and $V'$ are affine transformations, such as linear mappings or multilayer perceptrons.

In vector self-attentions, the representation of the self-attentive functions differs, like in Equation (15).

$$\mathbf{o}_i = \sum_{\mathbf{x}_j \in \mathcal{X}(\mathbf{i})} \psi\big(\alpha\big(\varphi(\boldsymbol{q}_i, \boldsymbol{k}_j)\big) + \boldsymbol{pos}\big) \odot (\boldsymbol{v}'_j + \boldsymbol{pos}) \tag{15}$$

$\mathcal{X}(\mathbf{i})$ is the set of vertices representing the vertices adjacent to $\mathbf{x_i}$, and $\varphi$ is a correlation function, e.g., $+$, $-$, $\times$, or $\odot$. $\alpha$ is a mapping function that maps a self-attentive vector to a self-attentive feature, e.g., MLP.

For the 3D mesh model, the 3D mesh transformer layer used in this paper is based on the vector transformer design. The obtained self-attention function is as follows.

$$\mathbf{o}_i = \sum_{\mathbf{x}_j \in \mathcal{X}(\mathbf{i})} \psi\big(\alpha\big(\boldsymbol{q}_i \odot \boldsymbol{k}_j\big) + \boldsymbol{pos}\big) \odot (\boldsymbol{v}'_j + \boldsymbol{pos}) \tag{16}$$

$\mathcal{X}(\mathbf{i})$ is the set representing the vertices adjacent to $\mathbf{x_i}$ (in this experiment it is the nearest neighbor k vertices), and the local self-attention mechanism is applied in the local neighborhood around each vertex. The correlation function $\varphi$ is chosen as the Hadamard product, and the mapping $\alpha$ function is an MLP with two linear layers and a Relu layer.

### 3.2.2. Learnable Position Encoding

In the self-attention, positional encoding can mark the relative position relationship between elements and introduce more spatial information. Therefore, positional encoding is often introduced to enhance the model effect in natural language processing or 2D image processing. In the 3D model, its original 3D coordinate information can reflect the position relationship between elements, in order to eliminate the influence of the coordinate system and coordinate scale on the results. We introduce a learnable position encoding method.

$$\boldsymbol{pos} = \beta\big(\boldsymbol{p}_i - \boldsymbol{p}_j\big) \tag{17}$$

$\boldsymbol{p}_i$ and $\boldsymbol{p}_j$ denote the 3D coordinates corresponding to the vertices $v_i$ and $v_j$ of the 3D mesh model. $\beta$ is an MLP with two linear layers and a Relu layer.

After processing by the transformer layer, new semantic features are obtained after performing local feature fusion. To reduce the number of parameters, a down layer is introduced to sample tokens. First, the input is sampled, and for processing convenience, the sampling function here is the farthest point sampling. Since some feature information is lost after sampling, the KNN method is introduced. The sampled tokens that aggregate the feature information of their K-neighborhoods are obtained by an MLP layer consisting of normalization and Relu and a K-neighborhood local maximum pooling layer.

### 3.2.3. Transformer down Layer

To extract the low-level features into high-level features and reduce the feature dimension, we design the transformer down layer. The transformer down layer is shown in Figure 3b. Where X represents the input feature vector and P represents the input vertex set. $X_{new}$ represents the output feature vector and $P_{new}$ represents the output feature vector. First, by sampling, we select a well-distributed subset of vertices $P_{new}$, $P_{new} \in P$. To pool the feature vector X associated with P onto the feature vector $X_{new}$ associated with $P_{new}$, we use KNN for P and then max-pool each point in P from the K neighboring points in P.

### 3.2.4. Transformer up Layer

For the semantic segmentation task, we combine the main modules of MixFormer with the U-net network. To decode the feature information extracted by the network, we designed the transformer up layer to map features from the set of downsampled input points P to its superset $P_{new}$. The transformer up layer is shown in Figure 3c. For this purpose, each input point feature is processed by a linear layer, followed by batch

normalization and Relu, and then the features are mapped to a higher resolution point set $P_{new}$ by trilinear interpolation. Finally, the interpolated features from the previous decoder level are provided with the corresponding encoder-level feature summaries by a jump connection.

### 3.3. Time Complexity Analysis

At the end of this chapter, we will discuss the time complexity of the methods in this paper.

In the mesh CNN block, we use Equation (7) to perform the similarity measure between the 3D convolution kernel and the local surfaces, which is calculated only with respect to the number of points in the local surfaces, and the number of points in the local surfaces is a custom constant K, so the similarity calculation time is a constant $t$. To obtain a specified number k of standard surfaces as 3D convolution kernels by clustering, all n local surfaces need to be traversed to calculate similarity. The time complexity is $O(I * n * k * t)$. In order to perform 3D convolution operation on a 3D mesh model using 3D convolution kernels, each convolution kernel needs to calculate similarity with all local surfaces, so the time complexity of convolution operation is $O(n * k * t)$. In summary, without considering the operations of the downsampling and local surface division, the time complexity of the 3D convolution module is $O(I * n * k * t + n * k * t)$, where I, k, and t can be regarded as constants, so the time complexity can be simplified to $O(n)$.

In the mesh transformer block, the time complexity is mainly focused on the operations of the self-attention. For the self-attention operation of layer L, the input token number is $N/4^L$, and the feature dimension d is $64*2^{L-1}$. Therefore, the time complexity of self-attention is $O((N/4^L)^2 d)$, where d and $4^L$ can be regarded as constants, so the time complexity can be simplified to $O(N^2)$.

## 4. Results and Discussion

In this section, we verify the effectiveness of our method in two applications: 3D mesh model classification and surface semantic segmentation. The network design details of the experiments are shown in Figure 4. We also have done ablation experiments for the sampling method, for the related function, and for the position encoding. We also tried different parameters in the mesh transformer block to explore the effect of the number of transformer layers and transformer output layer dimension on the experimental results.

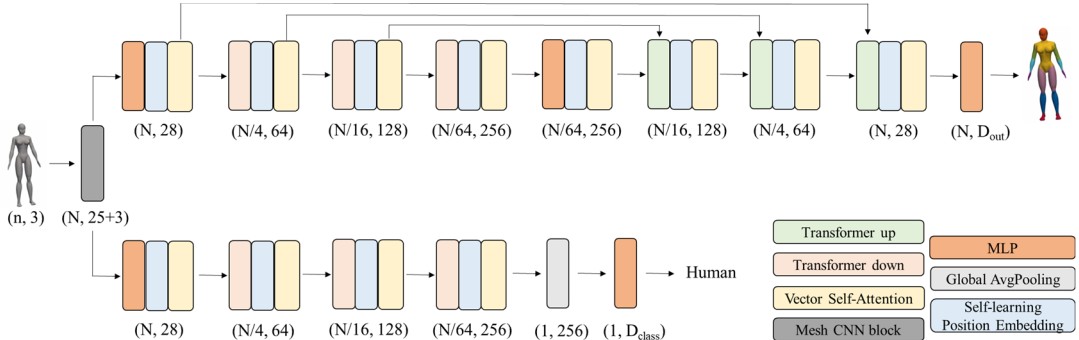

**Figure 4.** MixFormer segmentation network (**up**) and classification network (**down**).

The experimental host configuration is NVIDIA GeForce RTX 2080 Ti graphics card, Intel(R) Core(TM) i5-9600 CPU processor, and 32G RAM.

### 4.1. Model Parameters

The network model is built based on the Pytorch framework with Pytorch version 1.8.1. The 3D mesh model data is first convolved by the 3D mesh convolution block, and the convolved surface window is taken with the vertex as the center, and the neighborhood

range is 152 vertices close to the vertex. The vertices with the convolved surface information aggregated to the neighborhood are sampled by Poisson disk to take 10% of the points, which are then mapped to the feature space by the MLP layer and input to the 3D mesh transformer block.

*4.2. Classification Experiments*

4.2.1. Classification Experiments Based on SHREC15 Dataset

The SHREC15 dataset is derived from SHREC11 and SHREC14 and contains both rigid and non-rigid models. The SHREC15 dataset contains 50 categories with 24 models in each category, for a total of 1200 3D mesh models. The maximum mesh resolution in the dataset is 60,210 vertices, and the average mesh resolution is 21,141 vertices. In the ratio of 3:1, 18 3D mesh models are randomly selected from each category as the training set and the rest as the test set.

In order to verify the classification effect of MixFormer proposed in this paper, this paper compares the traditional manual feature-based classification method SPH with the 3D convolution-based classification methods MeshNet and MeshCNN. The accuracy and parametric number comparison of each method for classification in the SHREC15 dataset are given in Table 1.

**Table 1.** Classification accuracy of different methods on SHREC15.

| Method | Class Accuracy/% | | | | | | | | | | | Accuracy/% |
|---|---|---|---|---|---|---|---|---|---|---|---|---|
| | Alien | Ants | Cat | Dog1 | Dog2 | Man | Shark | Santa | Pliers | Glasses | Camel | |
| SPH | 87.4 | 86.2 | 90.4 | 89.3 | 86.7 | 89.1 | 90.2 | 89.4 | 87.1 | 89.9 | 88.1 | 88.2 |
| MeshNet | 89.5 | 89.6 | 89.6 | 91.4 | 90.5 | 90.8 | 90.1 | 89.8 | 88.0 | 91.4 | 90.3 | 90.4 |
| MeshCNN | 91.2 | 91.4 | 92.1 | 90.2 | 93.7 | 91.6 | 92.7 | 90.5 | 91.8 | 90.4 | 90.3 | 91.7 |
| Ours | 100 | 100 | 100 | 87.5 | 100 | 100 | 100 | 100 | 100 | 100 | 75.0 | 96.7 |

Due to the large number of classes included in SHREC15, only the class classification accuracies and the average accuracy of each method for all real columns for 11 of these models are shown in Table 1. It can be seen that the classification effect of the MixFormer proposed in this paper is better than other comparison methods, and the classification accuracy is 96.7%, which is 5 percentage points better than the optimal method MeshCNN among the comparison methods. This shows that the MixFormer proposed in this paper does have a good ability to learn features on 3D grid models and can achieve good classification results without deep network architecture

Table 2 compares the time and space complexity of the methods in this paper with those of other classification methods. The column Parmas shows the total number of parameters of the network, and the column FLOPs shows the number of floating point operations performed on each input sample, representing the spatial and temporal complexity, respectively. By learning local feature information through the front 3D grid convolution module and establishing dependencies between features through the 3D grid transformer, MixFormer has a stronger feature learning capability, so good learning results can be achieved without stacking multiple layers and with less computational overhead.

**Table 2.** Comparison of the classification accuracy of different pooling sampling methods.

| Method | Parmas/M | FLOPs/10$^9$ | Accuracy/% |
|---|---|---|---|
| SPH | 2.4 | 4.4 | 88.2 |
| MeshNet | 4.3 | 5.1 | 90.4 |
| MeshCNN | 1.3 | 5.0 | 91.7 |
| Ours | 0.8 | 1.2 | 96.7 |

### 4.2.2. Classification Results Based on the Manifold40 Dataset

Manifold40 [46] is derived from the ModelNet40 dataset, which is a new dataset contributed by Hu et al. after repairing the models in ModelNet40 to tight manifolds. In the Manifold40 dataset, 12,300 models with 40 categories are included. Manifold40 is more challenging due to the reconstruction error and simplification distortion in the Manifolds dataset.

To further validate the classification effect of the proposed MixFormer proposed in this paper, experiment was also conducted on the Manifold40 dataset and compared with the vertex-based methods: PointNet++ and PCT, and the mesh feature-based methods: MeshNet, MeshWalker, and SubdivNet. The accuracy of the classification of each method on the Manifold40 dataset is given in Table 3.

**Table 3.** Classification accuracy of different methods on Manifold40.

| Method | Accuracy/% |
| :---: | :---: |
| PointNet++ [47] | 87.9 |
| PCT [20] | 92.4 |
| MeshNet [16] | 88.4 |
| MeshWalker [48] | 90.5 |
| SubdivNet [46] | 91.5 |
| Ours | 93.6 |

From the comparison of the point cloud methods, it can be seen that the transformer-based PCT method works better than the PointNet++ based on multilayer convolutional networks, which implies that it may be important to obtain the dependencies of global features by introducing the transformer structure. From the comparison applied to the 3D grid model, it can be seen that the method in this paper improves by 2.1 percentage points over the SubdivNet method, which is the best among the compared methods, further indicating that the self-supervised convolutional architecture proposed by the method in this paper can learn the 3D mesh model features better.

### 4.3. Semantic Segmentation Experiments

To validate the feature learning effect of the proposed MixFormer in this paper, we used 370 models from SCAPE, FAUST, MIT, and Adobe Fuse as training data and 18 models from the human category in the SHREC07 dataset as test data. All models were segmented by Maron [49] et al. into eight categories of labels: head, torso, thigh, forearm, hand thigh, calf, and foot.

We combined the main module of MixFormer with the U-Net network to use the acquired model features for downstream tasks, such as semantic segmentation. Limited by the memory size of the graphics card, we first downsampled the model to 1024 local surfaces during the training process, and then predicted the human model. The semantic segmentation effect is shown in Figure 5.

From the segmentation visualization results, it can be seen that the proposed Mix-Former has good feature extraction ability and the ability to establish inter-feature dependencies, and can clearly delineate all parts of the human body and achieve good segmentation results. On the sampled model, mIoU is 0.849. However, some errors do exist at the joint demarcation line, which may be due to the fact that the marker of the model is not particularly fine and the joint demarcation line is difficult to define from the semantic point of view.

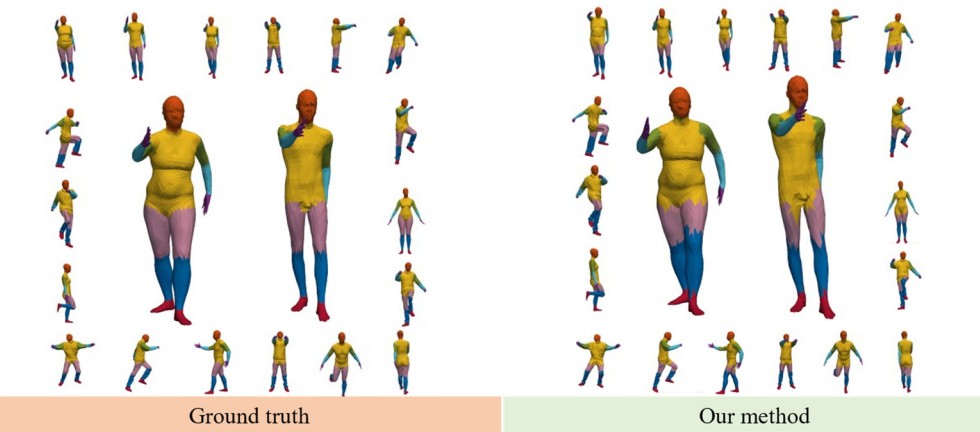

| Ground truth | Our method |

**Figure 5.** 3D human model torso structure semantic segmentation effect visualization (each part of the human body is marked with the following colors. Head: orange, torso: yellow, arms: green, forearms: light blue, hands: purple, thighs: root color, calves: dark blue, and feet: red).

### *4.4. Ablation Experiments*

To verify the effectiveness of the sampling method in MixFormer and the feature extraction method in transformer block, the following sets of ablation experiments are done for the sampling method, the correlation function, and the position encoding, respectively.

### 4.4.1. Sampling Method Validity

For the sampling methods, we selected random point sampling, farthest point sampling, and Poisson disk sampling for comparison. The experimental results are shown in Table 4. Compared with the other sampling methods, the sampling points selected by the Poisson disk can better reflect the surface contour characteristics of the original 3D mesh model. As far as the classification results are concerned, the Poisson disk sampling is used for data preprocessing to obtain better classification results for the classification network, 96.7%, which is 3.6 percentage points higher than the optimal method of farthest point sampling among the compared methods.

**Table 4.** Classification accuracy of different sampling methods.

| Method | Accuracy/% |
|---|---|
| Random sampling | 92.76 |
| Farthest point sampling | 93.17 |
| Poisson disc sampling | 96.76 |

### 4.4.2. Related Function

In the scalar transformer, the similarity is often calculated by dot product operation. In this paper, we adopt a vector transformer, which uses addition, subtraction, Hadamard product, or splicing as the correlation function, and obtains the vector output by the correlation function $\varphi$, which is used to measure the similarity between vectors. The experimental results in Table 5 show that compared with the traditional dot product operation of the scalar transformer, better classification results are often achieved by using the vector transformer, which indicates that the vector transformer may be more suitable than the scalar transformer for extracting the spatial features of the 3D mesh model. When the Hadamard product is used as the correlation function, a good classification result can be obtained with a relatively low number of parameters and arithmetic power consumption.

**Table 5.** Classification accuracy of different correlation functions.

| Related Function $\varphi$ | Parmas/M | FLOPs/$10^9$ | Accuracy/% |
|---|---|---|---|
| Add | 0.75 | 1.02 | 93.17 |
| Subtraction | 0.75 | 1.02 | 96.33 |
| Had. product | 0.75 | 1.02 | 96.76 |
| Concatenation | 0.80 | 1.24 | 94.33 |
| Dot product | 0.84 | 2.42 | 93.28 |

### 4.4.3. Position Encoding

For location coding, the following experiments were also conducted in this paper to compare several different location coding methods. As can be seen in Table 6, the classification accuracy of the model is 92.79% when no location encoding is introduced. When the conventional relative location encoding is introduced, the classification accuracy of the model improves to 94.08%, indicating that location encoding can indeed model the location relationship between tokens and introduce spatial information. The classification accuracy is further hinted at when we introduce the learnable location encoding, indicating that the learnable location encoding method can indeed better model the relative location and dependency relationships among tokens. To further explore the sensitivity of each part to location encoding, we acted location encoding on the attention part and feature part separately, and the experimental results showed that the attention part was more sensitive to location encoding. In this network, the learnable location encoding acting on the attention part and feature part together can achieve the optimal feature learning effect of 96.76%, which is nearly 4% higher than that without the introduction of location encoding.

**Table 6.** Classification accuracy with different Position encoding.

| Position Encoding | Parmas/M | FLOPs/$10^9$ | Accuracy/% |
|---|---|---|---|
| None | 0.64 | 1.02 | 92.79 |
| Absolute | 0.68 | 1.02 | 94.08 |
| Relative for attention | 0.75 | 1.02 | 95.44 |
| Relative for feature | 0.75 | 1.02 | 94.17 |
| Relative for both | 0.75 | 1.02 | 96.76 |

### 4.4.4. Number of Mesh Transformer Blocks

For the number of mesh transformer blocks, this paper tries different numbers of transformer blocks to explore their effects on the experimental results. From Table 7, we can see that the model performs best when the number of blocks is 2. When there are fewer transformer blocks, the network may not have enough deep-learning features. When there are more transformer blocks, the number of tokens input to the deep transformer block decreases and the token semantic information is blurred, and too many parameters are also introduced, which may reduce the network performance.

**Table 7.** Effect of the number of mesh transformer blocks on the classification accuracy.

| Number of Transformer Blocks | Accuracy/% |
|---|---|
| 1 | 92.7 |
| 2 | 96.7 |
| 3 | 87.4 |
| 4 | 75.2 |

### 4.4.5. Feature Dimensions of the Output Layer

We also tried different output layer feature dimensions to explore their effects on the model performance, and the experimental results are shown in Table 8. The network performance is optimal when the output layer feature dimension is 256.

**Table 8.** Classification accuracy with different dimensions of the output layer.

| Output Layer Dimension | Accuracy/% |
| --- | --- |
| 128 | 93.4 |
| 256 | 96.7 |
| 512 | 92.6 |
| 1024 | 87.3 |

## 5. Conclusions

Experiments show that the proposed MixFormer in this paper can achieve 96.7% classification accuracy on the dataset SHREC15, which is better than the rest of the methods applied to 3D mesh models, demonstrating the classification capability of the network. Moreover, we made a simple attempt at its semantic segmentation effect on the 3D mesh model to further demonstrate the feature learning capability of the network through a downstream task. Finally, ablation experiments were also performed to explore the effect of various factors on the network performance. In subsequent studies, its application to the semantic segmentation task of 3D mesh models or the introduction of a transformer decoder module for tasks such as unsupervised model generation can be further investigated in depth.

**Author Contributions:** Conceptualization, L.H., J.Z. and Y.C.; methodology, L.H., J.Z. and Y.C. All authors have read and agreed to the published version of the manuscript.

**Funding:** Supported by General Program of National Natural Science Foundation of China (No. 62071260 and No. 62006131). Supported by General Program of National Natural Science Foundation of Zhejiang Province (No. LZ22F020001).

**Data Availability Statement:** Data are contained within the article.

**Conflicts of Interest:** The author declares no conflict of interest.

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
