# Peer review of "MixFormer: A Self-Attentive Convolutional Network for 3D Mesh Object Recognition"

_algorithms, doi:10.3390/a16030171_

Round 1

Author Response

Thank you for your encouragement and affirmation. Thank you also for taking my articles seriously and finding the sparkle in my poorly written articles. Thank you.

Reviewer 2 Report

This paper proposes a self-attentive convolutional network MixFormer applied to 3D mesh models,and the expriments validate the superior performance. However, some minor issues below should be addressed in the resubmitted manuscript.  (1) Are the parameters used by the authors under different experimental settings? The experimental results obtained by using different parameters should be displayed in the manuscript. (2) There are some classification works on two dimensional images, such as Adaptive noise dictionary construction via IRRPCA for face recognition, Joint optimal transport with convex regularization for robust image classification,the differences of this work with them should be discussed, and  the  analysis of challenges to 3D data should be presented in the manuscript. (3) The motivations to give this paper could be better presented, especially the contributions or novelties of this paper could be clearer. It suggests revising the contributions section and making these points clear and strong.

Reviewer 3 Report

In this paper, the authors present a 3D mesh recognition model based on transformers. The idea is interesting yet the results are significant compared to SOTA models. the authors should provide a clear justification of that. Putting the code on GitHub for fairness of the results. 

1. Please explain the main novelty compared to SOTA models in the introduction section.

2. What makes the global attention compared to the local one, please explain.

3. Many equations are provided in the paper. please concentrate on the most important ones.

4. Discuss the computation complexity of the model.

5. It would be interesting to contrast the results using another dataset.

6. Please provide also attention maps to understand the visual aspect of the model.

Reviewer 4 Report

This paper proposes an architecture, namely a self-attentive CNN for 3D mesh object recognition. Here are my comments.

  1. Urgency and importance of this study is not clearly declared; How is the evaluation on existing studies? Why it is should be improved?
  2. The dataset used in this study is not well explained; Which one is used for classification, and which one is used for semantic segmentation; Is it the same for both?
  3. Compare the results, show with some examples, in the classification task, especially between "Ours" and "SPH" or "MeshNet"
  4. It's a little confusing, does it matter if you mention there are 50 categories (page 10) and 8 classes (page 11).
  5. Calculate the complexity of algorithms proposed in this study; If possible, show with the proof, e.g. using computational time measured in the inference process.
  6. Measuring the performance in semantic segmentation task usually uses metric like IoU/Jaccard or Dice coefficient. It is not properly measured with just accuracy. Please, provide more comprehensive and proper metrices.
  7. Please, check again some texts such as "CHREC15" in section 4.2 (page 10); "Figure 5" in section 4.3 (page 11); etc.
  8. Overall, I think this paper lacks important illustrations that can make it easier for readers to understand the idea of your proposed method.

Round 2

Reviewer 3 Report

I do not have additional comments.